# Expression of Apoptosis-Related Biomarkers in Inflamed Nasal Sinus Epithelium of Patients with Chronic Rhinosinusitis with Nasal Polyps (CRSwNP)—Evaluation at mRNA and miRNA Levels

**DOI:** 10.3390/biomedicines10061400

**Published:** 2022-06-13

**Authors:** Monika Morawska-Kochman, Agnieszka Śmieszek, Klaudia Marcinkowska, Krzysztof Mariusz Marycz, Kamil Nelke, Krzysztof Zub, Tomasz Zatoński, Marek Bochnia

**Affiliations:** 1Department of Otolaryngology, Head and Neck Surgery, Wroclaw Medical University, Borowska 213, 50-556 Wroclaw, Poland; krzysztof.zub@umw.edu.pl (K.Z.); tomasz.zatonski@umw.edu.pl (T.Z.); bachniab04@gmail.com (M.B.); 2Department of Experimental Biology, The Faculty of Biology and Animal Science, Wroclaw University of Environmental and Life Sciences, Norwida 27B, 50-375 Wroclaw, Poland; agnieszka.smieszek@upwr.edu.pl (A.Ś.); klaudia.marcinkowska@upwr.edu.pl (K.M.); krzysztof.marycz@upwr.edu.pl (K.M.M.); 3Department of Maxillofacial Surgery, 4th Military Clinical Hospital, Weigla 5, 50-981 Wroclaw, Poland; kamil.nelke@gmail.com

**Keywords:** chronic rhinosinusitis, nasal polyps, apoptosis, pro-apoptotic factors, anti-apoptotic factors, mRNA, miRNA

## Abstract

In chronic upper respiratory tract diseases, increased cell proliferative activity is observed, which is coordinated by BCL-2 proteins and small non-coding RNAs. This study aimed to determine the expression of critical apoptosis markers at the mRNA and miRNA levels in patients with chronic rhinosinusitis with nasal polyps (CSRwNP). The study group consisted of ten patients with CSRwNP and ten healthy subjects. To detect in situ apoptosis in the maxillary sinus mucosa, TUNEL staining was performed. The expression of transcripts was determined by RT-qPCR and included the detection of markers associated with cell survival and apoptosis, i.e., BAX, p53, p21, CASP3, CASP9, c-MYC, CCND1, BRIC5, and APAF1. Levels of miR-17-5p, miR-145-5p, miR-146a-5p, and miR-203a-3p were also measured by RT-qPCR. The obtained results indicated increased apoptosis determined by a TUNEL assay in CSRwNP patients and accompanied by an increased expression of BAX, P21, P53, CASP3, CASP9, c-MYC, and APAF-1 transcripts and decreased mRNA levels of BCL-2 and BIRC5. Furthermore, the nasal sinus epithelium of patients with CSRwNP showed increased levels of miR-203a-3p while also showing a decreased expression of miR-17-5p and miR-145-5p. Our results showed that pro-apoptotic transcripts detected at mRNA and miRNA levels might be involved in the pathogenesis of chronic sinusitis with polyps. The identification of those key molecular mediators may be applicable for the specific diagnostic and/or development of targeted therapies for chronic sinusitis with polyps.

## 1. Introduction

Apoptosis plays an essential role in eliminating damaged cells and preventing uncontrolled cellular proliferation [1,2]. In addition, genetic and epigenetic factors stimulate inflammatory responses in various autoimmune disorders, including chronic rhinosinusitis [3].

Shih et al. showed that chronic rhinosinusitis (CRS), with and without nasal polyps, demonstrated a significant association with premorbid autoimmune diseases (i.e., ankylosing spondylitis, polymyositis, psoriasis, rheumatoid arthritis, sicca syndrome, and systemic lupus erythematosus). However, the literature on the subject is sparse [4,5].

CRS phenotypes have been defined with or without nasal polyposis, and subphenotypes include allergic fungal rhinosinusitis and CRS-related aspirin-exacerbated respiratory disease. In addition, patients with CRS can be classified into three endotypes based on the presence of type 1, type 2, or type 3 inflammation [6].

The eosinophil count in nasal mucosa has been proved helpful for genotyping and assessing disease severity [7]. Regardless of atopy, most Caucasian patients with CRSwNP in Western countries exhibit type-2 immune responses with a tendency toward comorbidities characterized by elevated IL-4, IL-5, IL-13, and local IgE production and profound tissue eosinophilia. Approximately half of the CRSwNP patients in China and other East Asian countries have eosinophilic inflammation with a typical immune response with a type 2 bias [8]. Non-eosinophilic CRSwNP often has a predominant type 1 or type 3 immune response [9]. Therefore, taken as a whole, Asians with CRSwNP have a lower frequency of the type 2 cytokine expression, less eosinophilic inflammation, and less asthma comorbidity compared with their Caucasian counterparts [10].

Tomassen et al. divided patients with CRS based on tissue immune markers in a phenotype-free approach. They identified patients with CRS associated with TH2- and eosinophil-driven inflammation, neutrophilic and proinflammatory cytokines, H17-TH22-related markers, and TH1, IFN-g markers. Their research focused on the relationship between endotypes and CRS phenotypes. CRSwNP endotype markers with nasal polyposis indicate the desirability of using biological medicines targeting these receptors (IgE, IL-4, IL-5, and IL-13), especially in the treatment of CRS with nasal polyps and asthma (IL-5). While significant progress has been made in characterizing endotypes and phenotypes in CRS, additional studies are needed to determine how biomarkers could help physicians in individualized clinical treatment, and our work attempts to identify other biomarkers of CRSwNP [11].

In chronic rhinosinusitis, the inflammatory process causes the stroma to swell and form polyps. Nasal polyps (NP) are benign changes that originate in the lining of the nasal sinuses [12].

It was observed that patients with nasal polyps are characterized by increased epithelial cell proliferation-induced inflammation associated with the activation of epithelial repair mechanisms [13,14,15]. Therefore, one of the hypotheses is that the formation of polyps is related to abnormalities in cell proliferation and the inhibition of apoptosis [16,17]. Moreover, increased proliferative cellular activity has been noted in chronic upper respiratory tract diseases, such as asthma and bronchitis [18]. 

The programmed cell death is a multifactorial process that can occur either by the intrinsic or extrinsic pathways, i.e., a mitochondria apoptotic cascade or the death receptor-mediated pathway. The intrinsic pathway is coordinated by proteins of the Bcl-2(B-cell leukaemia/lymphoma 2) family. The proteins can act as pro-apoptotic or anti-apoptotic factors. The critical regulators of apoptosis are both BAX and BCL-2. BAX is an apoptosis-promoting protein that counteracts the anti-apoptotic function of BCL-2 by binding to this molecule. The translocation of pro-apoptotic proteins, e.g., BAX., to the internal mitochondrial matrix releases cytochrome C, which binds to APAF-1 (the apoptotic protease activating factor 1), inducing apoptosome formation [19,20,21]. In addition, the activation of caspase-3 (CASP3) within the apoptosome is indicative of the irreversible stage of apoptosis [22,23]. The increased BAX/BCL-2 ratio frequently correlates with a higher expression of the P53 protein involved in inhibiting cell proliferation. Besides, this may increase cyclin-dependent kinase inhibitor P21 protein expression [24]. 

Small, non-coding RNAs, namely miRNAs (18–24 nucleotides), play a vital role in regulating apoptosis by intrinsic and extrinsic pathways [25]. These molecules act post-translationally, targeting pro-and anti-apoptotic genes [25,26,27]. It was found that an overexpressed miR-761 might alleviate chronic inflammation and induce the remodelling of nasal mucosa in CRS mice. The miR-761 targeted LCN2 (lipocalin 2), decreasing its expression and inactivating the LCN2/Twist1 signalling pathway, accompanied by reduced apoptosis and inflammation [28]. Moreover, Zhang et al., determining the miRNA profile, showed that miR-125b could be significantly up-regulated in eosinophilic CRSwNP [29]. Thus, investigating miRNA/mRNA associated with apoptosis and inflammation characteristics in CRSwNP can be essential in identifying new therapeutic targets.

To our knowledge, this work is the first study analyzing the expression of selected apoptotic markers at mRNA and miRNA levels that may have potential prognostic utility in CRSwNP or may lead to the development of novel targeted therapies.

## 2. Materials and Methods

### 2.1. Patients

The study group consisted of 10 patients with CRSwNP (*n* = 10), including 7 women and 3 men (mean age = 54.5 years; SD = ±14; min. = 27; max. = 69). In each case, symptoms of rhinosinusitis persisted for more than one year. Based on anamnesis and treatment history, the results from available endoscopic studies (a rigid endoscope with 00 and 300 directions of view, before and after decongestion of the mucosa), and craniofacial computer tomography, all patients were qualified according to EPOS 2012 guidelines 15 for functional endoscopic sinus surgery (FESS) in the ENT Department. One month before FESS, no steroids or antibiotics had been used. A fragment of the inflamed mucosa was taken from the maxillary sinus during the procedure. Histopathological examination always confirmed the diagnosis of chronic eosinophilic sinusitis. Eosinophilic sinusitis, in our work, was determined by histological evaluation of the number of eosinophils in the visual field, which, according to EPOS 2020, should be ten or more in the visual field [30]. 

The control group was made up of 10 generally healthy people with only various orthognathic defects (*n*=10; 4 women and 6 men) (mean age = 27 years; SD = ±3; min. = 23; max. = 34). The main exclusion criteria were any pathologies of sinuses. Patients who qualified for Lefort I level osteotomy were treated at the Department of Maxillofacial Surgery at the local medical university. During hospitalization, blood tests were performed, particularly to measure inflammatory markers (blood cell count, C-reactive protein [CRP], procalcitonin, and fibrinogen levels) and CT and nasal endoscopy before and after surgery. The characteristic of the subjects is included in Appendix A (Appendix A). Excess mucosa of the maxillary sinus, obtained each time, was examined. The cytokines levels measured with RT-qPCR, CRP, and white blood levels are shown in Appendix A (Appendix A).

Each patient and healthy control signed an individual consent form and decided freely to participate in the study. The study was conducted in compliance with the principles of the Declaration of Helsinki.

### 2.2. TUNEL-Assay

Apoptotic cells in tissue samples were determined with the TUNEL Apoptosis Detection Kit (Cat # ab206386, Abcam, Cambridge, UK). Obtained paraffin blocks were cut into slices with 4-µm thickness. The tissue samples were washed three times with HBSS and fixed in 4% paraformaldehyde (PFA). Paraffin-embedded tissues were stained using a TUNEL assay according to the manufacturer’s instructions. Slides were deparaffinized in xylene (Sigma Aldrich, Munich, Germany) and rehydrated in an ethanol series (concentration from 100% to 70%). Then, specimens were permeabilized using a Proteinase K solution, and endogenous peroxidases were inactivated with 3% H_2_O_2_. Tissues were equilibrated in a TdT Equilibration Buffer and labelled with a TdT Labeling Reaction Mixture for 1.5 h at room temperature. Subsequently, the specimens were covered with a blocking buffer, and then a conjugate solution was added and incubated for 30 min in a humidified chamber at room temperature. Subsequently, a DAB solution was added for 15 min. Specimens were counterstained with a methyl green counterstain solution, mounted with DPX mounting media (Aqua-Med, Łódź, Poland), and analyzed using epi-fluorescent microscopy (Zeiss, Axio Observer A.1). The signals obtained after staining were determined using ImageJ and Pixel Counter Plugin (version 1.6.0, U.S. National Institutes of Health, Bethesda, MD, USA). 

### 2.3. Quantitative Real-Time Reverse Transcription Polymerase Chain Reaction (RT-qPCR)

The expression of apoptosis-related markers was analyzed on the mRNA and microRNA levels using well-established protocols [31,32,33]. The tissue samples were washed three times with Hanks’ Balanced Salt Solution (HBSS) and cut into small pieces using a scalpel. For homogenization, we used 1 mL of Extrazol^®^ (Blirt DNA, Gdansk, Poland). The total RNA was isolated according to the protocols provided by the manufacturer by using the phenol-chloroform method described by Chomczyński and Sacchi [34]. Obtained RNA was diluted in 30 μL of DEPC-treated water. The quantity and purity of RNA were evaluated using a spectrophotometer (Epoch, Biotek, Bad Friedrichshall, Germany) at 260 and 280 nm wavelengths. Before reverse transcription, the total RNA (500 ng) was treated with DNase I (Primerdesign, BLIRT S.A, Gdansk, Poland) to remove genomic DNA. cDNA synthesis was carried out using a Tetro cDNA Synthesis Kit (Bioline Reagents Limited, London, UK). According to the manufacturer’s protocols and the previously established methods, the gDNA digestion and cDNA transcription were performed using a T100 Thermal Cycler (Bio-Rad, Hercules, CA, USA). Additionally, for the evaluation of the miRNA level, after digestion, cDNA was synthesised using 375 ng of total RNA with a Mir-X™ miRNA First-Strand Synthesis Kit (Takara Bio Europe, Saint-Germainen, Laye, France). The obtained matrices were used for quantitative PCR using the SensiFAST SYBR^®^&Fluorescein Kit (Bioline Reagents Ltd., London, UK). Each reaction was performed in the final volume of 10 µL, where 1 µL of cDNA was used, and the concentration of primers was 0.5 µM. Quantitative PCR was performed in CFX Connect Real-Time PCR Detection System (Bio-Rad, Hercules, CA, USA). The cycling conditions used for transcript detection were as follows: initial denaturation at 95 °C for 2 min, followed by 45 cycles at 95 °C for 15 s, annealing for 15 s, and elongation at 72 °C for 15 s with a single fluorescence measurement. Each RT-qPCR was performed with at least three technical repetitions. 

The accuracy of the PCR reaction was evaluated by an analysis of the dissociation curve of the amplicons. The melting curve was performed using a gradient program at 55–95 °C at a heating rate of 0.2 °C/s with continuous fluorescence measurement. The primer sequences are summarized in Appendix A (Appendix A).

The expression of genes was calculated using the RQ_MAX_ algorithm and converted into a log2 scale as published previously. The normalization of detected transcripts was made in relation to the housekeeping gene—GAPDH (glyceraldehyde 3-phosphatedehydrogenase)—for mRNA and U6snRNA (Takara Bio Europe, Saint-Germainen, Laye, France) for miRNA. 

### 2.4. Statistical Analysis

The obtained results are presented as the mean from at least three technical repetitions. Means are presented with standard deviation (±SD). Statistical comparison between the groups was determined using an unpaired Student’s *t*-test. The data were analyzed using GraphPad Prism 8 software (La Jolla, CA, USA). Differences with a probability of *p* < 0.05 were considered statistically significant. The data used for comparative statistics are available in Appendix A (Appendix A).

## 3. Results

### 3.1. Apoptosis In Situ

The analysis revealed induced apoptosis in the CRSwNP tissue samples. The apoptotic cells (stained brown) were counted after the assay. They were significantly increased compared with the healthy control (Figure 1).

### 3.2. The Level of Genes Associated with Apoptosis in CRSwNP Patients

The analysis of RT-qPCR expression revealed that tissue samples from CRSwNP patients exhibited a significantly higher mRNA expression of pro-apoptotic genes, including *BAX*, *P53*, *P21*, *c-MYC*, *CASP3*, *CASP9*, and *APAF1*. The increased expression of these transcripts correlated with lower expression of anti-apoptotic *BCL-2*, as well as with survivin (*BIRC5*) and cyclin D (*CCND1*) (Figure 2. The comparative analysis of the miRNA levels revealed the increased expression of miR-203a-3p and the decreased expression of miR-17-5p and miR-145-5p in patients with CRSwNP. Notably, we found no significant differences in the miR-146a-5p levels between healthy subjects and CRwNP patients (Figure 3). Observed patterns of mRNA and miRNA were also indicated in Figure 4.

## 4. Discussion

The pathogenesis of chronic rhinosinusitis with nasal polyps (CSRwNP) is complex, especially the molecular background. The regulation of apoptosis in inflammatory diseases by intracellular peptides and signalling proteins is not fully understood yet [3,35]. In this study, we have determined, for the first time, the mRNA and miRNA levels of markers associated with apoptosis and measured them in nasal sinus epithelium. 

We have tested the mRNA expression of anti-apoptotic BCL-2 and pro-apoptotic BAX. Fan et al. examined the eosinophilic apoptosis in the mucous membrane of patients with chronic allergic rhinosinusitis with an increased BAX expression. They observed no differences in the superficial and deep layers of the mucous membrane [36]. The role of BCL-2 and BAX in sinus mucosal inflammation was also studied using *Staphylococcus aureus*, confirming their importance in the bacteria-induced apoptosis of olfactory cells [37]. Tesfaigzi et al. observed that respiratory tract exposure to environmental allergens or toxins might cause mucous membrane damage, followed by an inflammatory response and a lower expression of BCL-2. Recovery means reducing the number of cells to the state before damage/inflammation, with a temporarily decreased expression of BCL-2 as an apoptosis inhibitor [38]. Cohen et al. examined decreased levels of BCL-2 in patients with bronchial asthma and apoptosis inhibition [18]. We also reported increased levels of *BAX* mRNA and reduced levels of *BCL-2* in the mucous membrane of patients with CRSwNP, which may confirm sinus mucosa damage, followed by an inflammatory process and apoptosis inhibition.

In this study, we have also tested the mRNA levels of *P53* in patients with CRSwNP and compared them with levels noted for healthy subjects. The tumour suppressor oncoprotein-p53, known as “the guardian of the genome,” regulates the gene transcription responsible for, e.g., DNA repair, cell cycle, ageing, and proliferation processes [39,40,41]. Cell cycle repression and apoptosis play a similar role in oncogenesis suppression. Due to protein antagonistic bifunctionality, a single cell action enables an accurate reaction to change environmental conditions [42]. Ingle et al. indicated that in benign epithelial lesions of the upper respiratory tract, an accumulation of the p53 protein was noted as occurring without gene mutation [39]. Our study found higher levels of *P53* transcripts in patients with CRSwNP, which corresponded with the increased proliferation of epithelial cells.

A higher expression of P53 in CRSwNP was reported in 1999 by Lavezzi et al. [43]. Garavello et al. noted the overexpression of P53 and increased apoptosis in mucous membranes with polyps compared to the healthy sinus mucous membranes with an unchanged expression of the p21 protein [44]. Chalastras et al. examined patients with chronic rhinosinusitis and reported an increased tendency toward proliferation and decreased apoptosis in epithelial cells and an increased expression of P53, and decreased levels of BCL-2 in hyperplastic polyps [45]. On the other hand, Küpper et al. reported a reduced expression of P53 in patients with CRSwNP [17]. As shown above, the expression of P53 in CRSwNP varies greatly. One reason may be that researchers used, as the study control, fragments of tissue collected from different anatomical locations (e.g., sinuses, the nasal cavity, and the inferior nasal turbine) in some experiments. Another reason for the disparity in the results might be other diagnostic methods [17,43,45]. When P53 protein accumulation is reported with immunohistochemical studies, it does not always correspond with its transcript levels. The accumulation of the P53 protein in healthy cells may be very discreet and difficult to estimate [39,45]. Therefore, we have also tested the influence of CRSwNP on *P21* mRNA levels in the nasal epithelium. The P21 is a multifunctional protein regulating the cell cycle and ensuring genome stability in response to various stimuli, including DNA damage [46]. We observed a higher expression of the *P53* and *P21* transcripts in patients with CRSwNP, but the P53/P21 ratio reflecting their co-expression was at the same level in both healthy subjects and patients affected by CRSwNP.

Elevated levels of *BAX*, *P53*, and *P21* genes, and reduced levels of *BCL-2*, evidenced increased mucosal apoptosis in patients with CRSwNP. Moreover, the apoptotic index determined in tissue based on in situ TUNEL staining correlated with the transcriptome profile determined.

According to Jung and Hermeking, c-MYC activation is correlated with an increased expression of CCND1 and a lower expression of BCL-2 and P21[47]. Thus, we also determined the mRNA levels of both c-MYC and CCND1 to establish their expression pattern in CRSwNP patients and noted increased transcript levels of *c-MYC* and lower mRNA expression of *CCND1*. We also detected a significantly higher expression of *APAF-1*, *CASP3*, and *CASP9*, contributing to the induction of apoptotic processes. APAF-1 is a cytosolic protein that activates caspase-9, a key factor in the apoptotic pathway of mitochondrial cells [48]. APAF-1 remains in its autoinhibited form in healthy cells, and CASP-3 and CASP-9 remain inactive [49]. Cho et al. reported no apparent difference in CASP3 expression levels between normal mucosa and mucosa with nasal polyps [50]. CRSwNP can be, however, compared to other inflammatory processes. For example, Hou et al. found increased expression levels of *P53*, *CASP3*, and *CASP9* and decreased expression levels of BCL-2 in patients with rheumatoid arthritis (RA) [51]. Our data corroborate this gene expression pattern and suggest a convergent molecular mechanism of apoptosis activation.

Our study also decided to test the mRNA levels of *BIRC5* (survivin), which acts as a prognostic biomarker in neoplastic diseases, indirectly inhibits apoptosis, and promotes cell proliferation. Qui et al. found that its overexpression, associated with immune response [52,53], might play a key role in nasal polyps’ development [53]. However, they examined mucosal samples from various locations (the turbinate, tissue surrounding polyps, and nasal polyps). Cho et al., who examined mucosa from the inferior turbinate, did not report a higher survivin expression in the healthy group and patients with nasal polyps [50]. Instead, we reported lower expression levels of BIRC5, which is characteristic of increased apoptosis. Due to its immunomodulatory effect, BIRC5 may be a potential therapeutic target in CRSwNP. However, this hypothesis requires further research. 

The main goal of CRS’s current research is to understand the disease’s etiopathology and investigate new pathways of information transmission between cells (including the role of RNA). Extracellular vesicles (EVs) are endocytic nano-vesicles released by cells and found in human body fluids, including secretions from the nasal mucosa. They contain both mRNA and microRNA (miRNA). Research by Cha et al. revealed that the expression of extracellular vesicles miRNA differed depending on the chronic phenotype of non-nasal sinusitis (CRSsNP) and chronic nasal polyposis sinusitis (CRSwNP). By transferring a miRNA from one cell to another, EVs can play a functional role in CRS development [54]. 

In our study, for the first time, we have also tested the levels of small non-coding RNA, i.e., or microRNAs (miRNAs), in chronically inflamed nasal sinus epithelium. The miRNAs play a crucial role in many pathways that regulate apoptosis, as in cellular proliferation, differentiation, and ageing. Moreover, these molecules regulate gene expression in human diseases, serving as valuable indicators of some pathological processes, including cancer progression [55,56] and inflammatory diseases [57,58]. Thus, the growing evidence indicates that the molecules may also serve as valuable biomarkers in diagnosing diseases with excessive apoptosis. For instance, Bhaumik et al. showed that the increased expression of miR-146a/b inhibited an excessive secretion of inflammatory cytokines [59]. In turn, Sun et al. analyzed miR-125-5p and miR-143/145 levels showing their function as potential biomarkers for ischemic stroke [60]. Tang et al. examined the expression level of miR-145-5p and found that it was decreased in the synovial tissues of RA patients [61].

Moreover, miRNA molecules are used in therapy for many diseases. For example, Zhao et al. treated patients with traumatic brain injury with miR-203 inhibitors and reduced neuronal apoptosis by inhibiting CASP3 activity and increasing BCL-2 expression [62]. Furthermore, Cimmino et al. demonstrated that miR-15a and miR-16-1 negatively regulated BCL-2 at the post-transcriptional level [63]. Moreover, it has been found that miRNAs (the miR-106b or miR-29 cluster) can regulate the activity of P53 and P21 proteins in the processes of DNA damage [64,65]. In turn, the study by Liu et al. reported an association between miR-125b, miR-133, miR-146a, and miR-203 with acute exacerbation risk, inflammation, and severity of the chronic obstructive pulmonary disease. In addition, they reported a positive correlation of those molecules with the levels of inflammatory cytokines (TNF-α, IL-1β, IL-6, IL-8, IL-17, and IL-23) [66]. Furthermore, Taganov et al. suggested that miRNAs could regulate cellular immune responses to particular pathogens. They found that miRNAs acted as potentially negative regulators of inflammation and showed increased expression levels of miR-132, miR-146, and miR-155 in monocytic cells as a response to bacterial infection [67]. The collected data indicate that miRNAs play an essential role in apoptosis regulation during inflammation. However, miRNA levels are variable, however tissue and cell-specific [68]. In our study, using validated RT-qPCR assays, we decided to measure levels of miR-17-5p, miR-145-5p, miR-146a-5p, and miR-203a-3p.

Our study showed no difference between levels of miR-146a-5p in the healthy group and CRSwNP, which can be explained by Bhaumik et al. [59], who suggested that a delayed miR-146a/b induction might be a compensatory response to restrain inflammation. So far, no studies have investigated the role of miRNAs in CRSwNP. Bearing in mind the obtained data, the miRNA-203a overexpression and lowered levels of miR-17-5p and miR-145-5p can promote increased apoptosis in CRSwNP. The miR-17 is a biomarker of chronic immune diseases and cardiovascular and neurodegenerative diseases [69]. On the other hand, miR-145-5p and miR-146a-5p exhibited anti-apoptotic roles [68,70,71]. The miR-203a-3p overexpression was previously associated with increased apoptosis [72,73]. Thus, we think that miR 203a-3p could be a potential therapeutic target for CRSwNP treatment, but this hypothesis requires further functional studies.

The advantage of our study relies on tissue sampling, which allows us to compare the levels between healthy subjects and CRSwNP patients. In our study, tissue samples were taken from the exact location, i.e., the maxillary sinus mucosa. However, the study may be limited by the small sample size, also from healthy participants. This group is, however, rarely accessible for examination.

## 5. Conclusions

In CRSwNP, the inflammation is triggered and perpetuated by cytokines produced by the structural and infiltrating cells. Thus, the proinflammatory cytokines became essential biomarkers allowing for the diagnosis and prognosis of the disease. However, little is known about the apoptotic factors and their significance in the pathogenesis of CRSwNP. In our study, in situ identification of apoptotic cells by TUNEL staining showed significantly increased apoptosis in sinus mucosa in patients with CRSwNP. Furthermore, the pro-apoptotic state in the tissue microenvironment also correlated with higher mRNA levels of pro-apoptotic markers, i.e., *BAX*, *P53*, *P21*, *CASP3*, *CASP9*, and *APAF-1*. In summary, by using mRNA screening and functional verification, and clinical evidence, our study demonstrated the potential of pro-apoptotic markers as promising prognostic biomarkers for CRSwNP.

The expression of an anti-apoptotic transcript, i.e., *BCL-2**,* and the pro-proliferative factor—*BIRC5*—decreased. Increased levels of the *BAX* transcripts and decreased *BCL-2* indicated increased apoptosis in the inflamed mucous membrane of patients with CRSwNP. A comparison of p53 and p21 in the healthy control and the study group suggested disturbances in apoptosis regulation. Moreover, the obtained studies showed, for the first time, increased levels of miR 203a-3p with decreased levels of miR 17-5p and miR-145-5p at the same time, suggesting that miR 203a-3p may be a potential therapeutic target for the treatment of CRSwNP. To the best of our knowledge, this is the first report investigating the biological functions of miR 203a-3p with miR 17-5p together and their clinical implications in CRSwNP.

For a precision medicine-based approach to patient treatment, a profound understanding of the pathology of this heterogeneous disease is crucial. This study may fundamentally influence CRSwNP research, particularly for identifying new biomarkers and future therapeutic targets, paving the way for new pharmacological interventions.

## Figures and Tables

**Figure 1 biomedicines-10-01400-f001:**
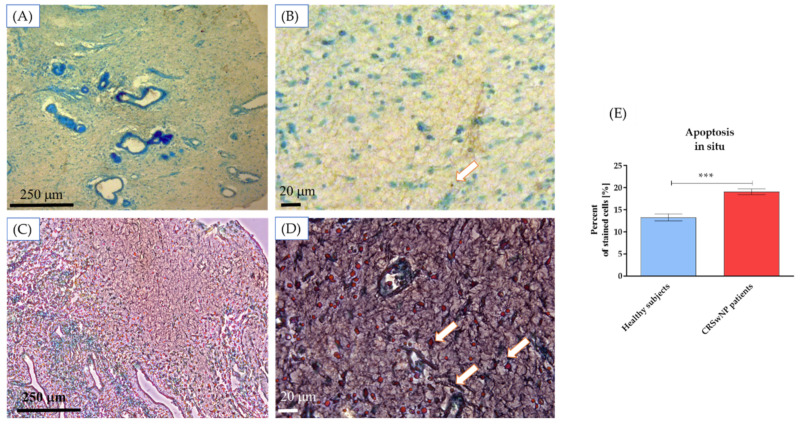
Apoptosis detection in tissue sections derived from healthy subjects (**A**,**B**) and CRSwNP patients (**C**,**D**). The statistical analysis was performed on representative images (healthy subjects *n* = 5 and CRNwNP *n* = 5). Scale bars, 250 and 20 µm are indicated in the microphotographs. The white arrows indicate apoptotic nuclei. The statistical analysis results show significantly increased apoptotic cells in tissues of CRSwNP patients (**E**). Statistically significant differences were noted at *p* < 0.001 (***).

**Figure 2 biomedicines-10-01400-f002:**
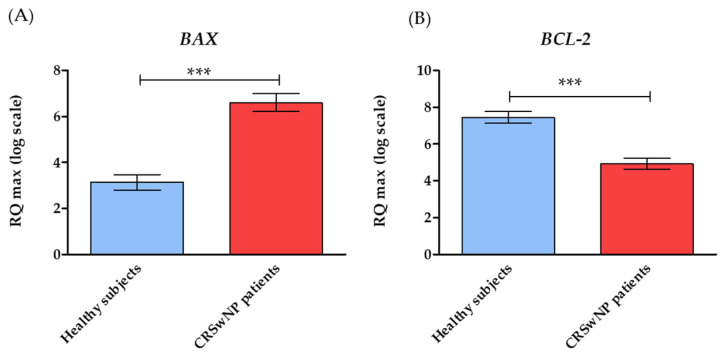
The expression profile of genes was associated with apoptosis and cell survival. Transcript levels were established for CRSwNP patients to the healthy subjects. The following markers at mRNA were detected: *BAX*—apoptosis regulator BCL2 Associated X (**A**); *BCL-2*—B-cell CLL/lymphoma 2 (**B**); *P53*—Tumor Protein P53 (**C**); *P21*—cyclin-dependent kinase inhibitor 1A (**D**); *BIRC5*—Baculoviral IAP repeat-containing protein 5 (**E**); *CASP3*—Caspase 3 (**F**); *CASP9*—Caspase 9 (**G**); *APAF1*—Apoptotic protease-activating factor 1 (**H**); CCND1—G1/S-specific cyclin-D1 (**I**); *cMYC*—Myc proto-oncogene protein (**J**). The average fold change of the target genes was determined in relation to the housekeeping gene, i.e., glyceraldehyde 3-phosphate dehydrogenase (GAPDH). Statistically, significant differences are indicated with an asterisk. Statistically significant differences were noted at *p* < 0.01 (**), and *p* < 0.001 (***).

**Figure 3 biomedicines-10-01400-f003:**
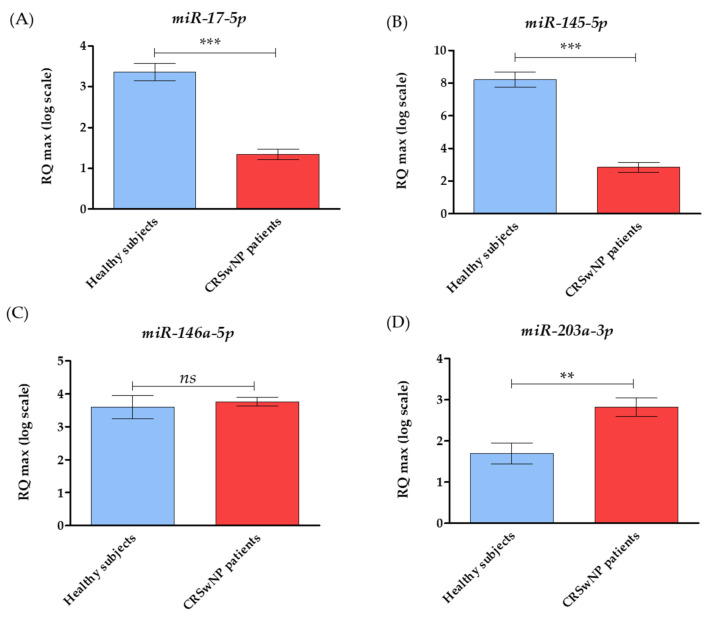
The miRNA levels were associated with apoptosis and cell survival. Transcript levels were established for CRSwNP patients and compared to levels determined in healthy subjects. The following miRNAs were measured: miR-17-5p (**A**); miR-145-5p (**B**); miR-146a-5p (**C**), and miR-203a-3p (**D**); the average fold change of the target genes was determined in relation to the housekeeping gene, i.e., small nuclear RNA U6 (snU6). Statistically, significant differences are indicated with an asterisk. Statistically significant differences were noted at *p* < 0.01 (**), and *p* < 0.001 (***).

**Figure 4 biomedicines-10-01400-f004:**
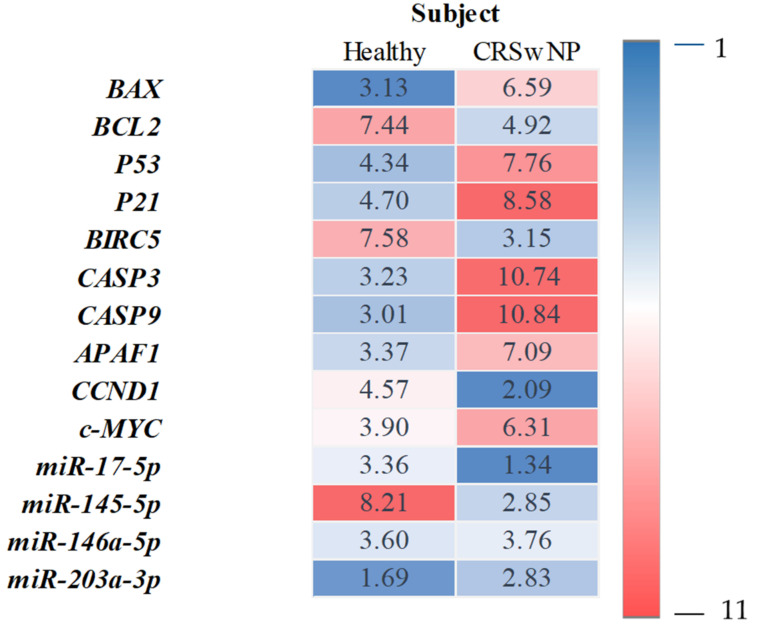
Summary of the obtained transcripts expression profiles. Heatmap showing the examined mRNA and miRNA levels in the healthy subjects (*n* = 10) and CRSwNP patients (*n* = 10). Each RT-qPCR was done with at least three technical repetitions. The heatmap shows the mean relative transcript amount obtained for healthy subjects and CRSwNP. Levels of down expression (blue) or overexpression (red) are shown on a log2 scale from the high to the low expression of each gene.

## Data Availability

https://www.researchsquare.com/article/rs-899520/v1; accessed on 24 September 2021.

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
