# Peer review of "Expression of Apoptosis-Related Biomarkers in Inflamed Nasal Sinus Epithelium of Patients with Chronic Rhinosinusitis with Nasal Polyps (CRSwNP)—Evaluation at mRNA and miRNA Levels"

_biomedicines, 2022, doi:10.3390/biomedicines10061400_

Round 1
Reviewer 1 Report
I appreciate the opportunity to review the manuscript for publication in MDPI Biomedicines.
The authors examined that pro-apoptotic transcripts detected at mRNA and miRNA levels might be involved in the pathogenesis of CRSwNP. However, I feel that this paper is premature and the present configuration may be not worthy of publication in the journal based on the editor's discretion.
The title should include a clear phenotype of CRS; i.e. chronic rhinosinusitis with nasal polyps (CSRwNP).
The authors tried to evaluate apoptosis processes of the sinonasal mucosa under inflamed conditions. The topics are interesting. However, the body of manuscript should be further focused on specific pathophysiological conditions of CRSwNP based on difference in each endotype as well as Type2 inflammation.
Most of references cited in the Introduction to draw attention on airway apoptosis are rather published in older days. They should be renovated.
The study group number (n=10) is rather small to bear statistics. The authors should describe how to determine the adequate sample number enough to draw reliable results based on previous risk ratios.
L84: “During the procedure, a fragment of the inflamed mucosa was taken from the maxillary sinus.”
I think either nasal polyps or ethmoidal mucosa is suitable for investigation.
L85: “Histopathological examination always confirmed the diagnosis of chronic eosinophilic sinusitis.”
The authors should specify the criteria including Eo cell density.
L133: “the SensiFAST SYBR®&Fluorescein Kit (Bioline Re-133 agents Ltd., London, United Kingdom)”
The kit is not suitable for detection of subtle differences. Gene-specific Taqman Probe should be considered.
The authors should display patients’ background more in detail using additional tables.
Figure 1: The quality of IHC in Figure 1 is very poor. Most of epithelial cells in (A) are exfoliated and damaged. There are countable numbers of spotted areas caused by artifacts. Overall, how can the authors reach the illustration of Fig. 1c with clear statistics.
It is well known that miRNA are contained and released through extracellular vesicles (EVs) of endocytic origin released by cells. They are found in human body fluids including nasal secretion. The authors should present further data focusing on the points.
Cha S, Seo EH, Lee SH, Kim KS, Oh CS, Moon JS, Kim JK. MicroRNA Expression in Extracellular Vesicles from Nasal Lavage Fluid in Chronic Rhinosinusitis. Biomedicines. 2021 Apr 26;9(5):471. doi: 10.3390/biomedicines9050471. PMID: 33925835; PMCID: PMC8145239.
Author Response
Responses to the Reviewer 1
Comment 1.
I appreciate the opportunity to review the manuscript for publication in MDPI Biomedicines.
The authors examined that pro-apoptotic transcripts detected at mRNA and miRNA levels might be involved in the pathogenesis of CRSwNP. However, I feel that this paper is premature and the present configuration maybe not worthy of publication in the journal based on the editor's discretion.
Answer 1.
Dear Reviewer, thank you for your opinion. The presented results are original and have the aspect of novelty thus, we are convinced that the paper will be interesting and valuable for the Readers of Biomedicine.
Comment 2.
The title should include a clear phenotype of CRS; i.e. chronic rhinosinusitis with nasal polyps (CSRwNP).
Answer 2.
We are grateful for the suggestion. The title has been corrected accordingly. The current title of the paper is Expression of apoptosis-related biomarkers in inflamed nasal sinus epithelium of patients with chronic rhinosinusitis with nasal polyps (CRSwNP) – evaluation at mRNA and miRNA levels.
Comment 3.
The authors tried to evaluate apoptosis processes of the sinonasal mucosa under inflamed conditions. The topics are interesting. However, the body of manuscript should be further focused on specific pathophysiological conditions of CRSwNP based on differences in each endotype as well as Type2 inflammation.
Answer 3.
Dear reviewer
Thank you, for your comments. We have completed the introduction with information on eosinophilic inflammation:
“Regardless of atopy, most Caucasian patients with CRSwNP in Western countries exhibit type-2 immune responses with a tendency toward comorbidities characterized by elevated IL-4, IL-5, IL-13 and local IgE production and profound tissue eosinophilia. Approximately half of the CRSwNP patients in China and other East Asian countries have eosinophilic inflammation with a typical immune response with a type 2 bias. (Zhang Y, Gevaert E, Luo H, et al. Chronic rhinosinusitis in Asia. J Allergy Clin Immunol. 2017; 140:1230–1239.) Non-eosinophilic CRSwNP often has a predominant type 1 or type 3 immune response. (Wang H, Li ZY, Jiang WX, et al. The activation and function of IL-36γ in neutrophilic inflammation in chronic rhinosinusitis. J Allergy Clin Immunol. 2018; 141:1646–1658.)Therefore, taken as a whole, Asians with CRSwNP have a lower frequency of type 2 cytokine expression, less eosinophilic inflammation, and less asthma comorbidity compared with their Caucasian counterparts.” (Cao, Ping-Ping et al. “Pathophysiologic mechanisms of chronic rhinosinusitis and their roles in emerging disease endotypes.” Annals of allergy, asthma & immunology : official publication of the American College of Allergy, Asthma, & Immunology vol. 122,1 (2019): 33-40. doi:10.1016/j.anai.2018.10.014)
Comment 4.
Most of references cited in the Introduction to draw attention on airway apoptosis are rather published in older days. They should be renovated.
Answer 4.
The references were updated and improved.
Comment 5.
The study group number (n=10) is rather small to bear statistics. The authors should describe how to determine the adequate sample number enough to draw reliable results based on previous risk ratios. – pytanie do statystyka
Answer 5.
In order to evaluate the test repeatability performance of the measurements, we used Stain’s formula.
The number of repetitions required to obtain significance (n) was measured for each assay. The t value at 9 degrees of freedom was taken from the Student distribution tables and was equal to 2.262. As a result, the calculated value of the minimum sample size (n) was lower than the assumed number of subjects used in the study (n=10), and the results were considered reproducible. The results of the calculations are presented in the Table below.
As authors, we agree that the group is small, but we do hope the number of markers will compensate for this. We wanted the number of respondents to correspond to the control group. However, it is not easy to obtain histological material from healthy people. We think that it will indicate further directions for developing research on CRS in NP.
We also patterned our work after papers that were published using a small study group, e.g.
Cheng, J., Chen, J., Zhao, Y. et al. MicroRNA-761 suppresses remodeling of nasal mucosa and epithelial–mesenchymal transition in mice with chronic rhinosinusitis through LCN2. Stem Cell Res Ther 11, 151 (2020). https://doi.org/10.1186/s13287-020-01598-7 -
Viksne RJ, Sumeraga G, Pilmane M. Characterization of Cytokines and Proliferation Marker Ki67 in Chronic Rhinosinusitis with Nasal Polyps: A Pilot Study. Medicina (Kaunas). 2021 Jun 11;57(6):607. doi: 10.3390/medicina57060607. PMID: 34208325; PMCID: PMC8231174.
Comment 6.
L84: “During the procedure, a fragment of the polypoid changed inflamed mucosa was taken from the maxillary sinus.” I think either nasal polyps or ethmoidal mucosa is suitable for investigation.
Answer 6.
Nasal polyps are derived from all sinuses, not only ethmoid sinus, as the nasal mucosa does normally show less ability to form polyps, and therefore, the term nasal polyps are misleading. A severe case of type 2 mucosal disease involving all the sinuses, therefore, the mucosa was collected from the polypoid changed, inflamed maxillary sinus. The control group had taken the mucosa from exactly the same location, so it was referred to the same studies on healthy people who had no polyps and no open ethmoid sinuses.
Comment 7.
L85: “Histopathological examination always confirmed the diagnosis of chronic eosinophilic sinusitis.”The authors should specify the criteria including Eo cell density.
Answer 7.
Eosinophilic sinusitis was determined by histological evaluation of the number of eosinophils in the visual field, which according to EPOS 2020 should be 10 or more in the visual field.
Fokkens WJ, Lund VJ, Hopkins C, et al. Executive summary of EPOS 2020 including integrated care pathways. Rhinology. 2020 Apr;58(2):82-111. DOI: 10.4193/rhin20.601. PMID: 32226949.
Comment 8.
L133: “the SensiFAST SYBR®&Fluorescein Kit (Bioline Reagents Ltd., London, United Kingdom)” The kit is not suitable for detection of subtle differences. Gene-specific Taqman Probe should be considered.
Answer 8.
Dear Reviewer, the RT-qPCR specificity, efficiency, and sensitivity were assured. The temperature of primer hybridization (annealing temperature) was optimized to achieve 95% of assay efficiency. The materials and methods section mentioned that the products were verified based on their melting temperature. During the reaction, no-reverse transcriptase (no-RT) and no-template controls (NTC) were used to discriminate between specific and off-target or non-specific products. We agree that the Taqman technology is a more specific assay, however, for the research purpose, SybrGreen-based detection of transcripts is also widely applied. The fluorescein was used to provide an internal reference to which the reporter dye(Sybr Green) signal can be normalized during data analysis,
Comment 9.
The authors should display patients’ background more in detail using additional tables.
Answer 9.
Additional data on the tested group will be added as supplementary material indicating pro-inflammatory phenotype of CRSwNP patients.
Comment 10.
Figure 1: The quality of IHC in Figure 1 is very poor. Most of epithelial cells in (A) are exfoliated and damaged. There are countable numbers of spotted areas caused by artifacts. Overall, how can the authors reach the illustration of Fig. 1c with clear statistics.
Answer 10.
Ten fields with apoptotic cells were observed in specimens at 100× magnification. We used the ImageJ software to determine the number of cells in each field based on staining intensity. The data were expressed as the mean percentage of apoptotic cells per microscopic field.
Comment 11.
It is well known that miRNA are contained and released through extracellular vesicles (EVs) of endocytic origin released by cells. They are found in human body fluids including nasal secretion. The authors should present further data focusing on the points.
Answer 11
Thank you for pointing out the issue, however our aim was to determine the microRNA levels within the tissue as they serve also as tissue specific markers. Bearing in mind the reference provided by the Reviewer we improved the discussion section in order to deliberate the issue of miRNA as potential biomarkers in liquid biopsies and tissue (in situ).
The main goal of CRS' current research is to understand its etiopathology of disease and investigate new pathways of information transmission between cells (including the role of RNA). Extracellular vesicles (EV) are endocytic nano-vesicles released by cells and found in human body fluids, including secretions from the nasal mucosa. They contain both mRNA and microRNA (miRNA). Research by Cha et al. Revealed that the expression of extracellular vesicles miRNA differs depending on the chronic phenotype of non-nasal sinusitis (CRSsNP) and chronic nasal polyposis sinusitis (CRSwNP). By transferring a miRNA from one cell to another, EVs can play a functional role in CRS development.
Cha S, Seo EH, Lee SH, Kim KS, Oh CS, Moon JS, Kim JK. MicroRNA Expression in Extracellular Vesicles from Nasal Lavage Fluid in Chronic Rhinosinusitis. Biomedicines. 2021 Apr 26;9(5):471. doi: 10.3390/biomedicines9050471. PMID: 33925835; PMCID: PMC8145239.

Reviewer 2 Report
I would like to thank the authors for all of there work on this important topic. This contribution helps fill in a gap in our understanding of the Inflamatory processes invloved in chronic rhinosinusitis, however there are a few areas within the paper that would bennifit from clarification and more information if possible.
line 38 suggest that chronic rhinosinusitis is an autoimmune disorder, and while there is significant evidence that autoimmune disorders contribute to the inflammatory processes involved, CRS id not considered an autoimmune disorder.
With the introduction of IG E monocolonal antibodies as a treatment for CRS, there has been significant research and discussion focused on distinguishing genotype from phenotype in Chronic Rhinosinusitis especially when considering medical therapy. Throughout the paper you target the phenotype CRSwNP yet utilize genetic and molecular markers to define causes of inflammation involved. While Type II inflammation is most often involved in CRSwNP in wester populations, it is the exact opposite in Asian Populations where type I inflammation is more typically involved in polyp formation. In line 86 you mention that you identified all patients had confirmed eosinophilic sinusitis, can you clarify that the genotype of each or the 10 patients with CRSwNP was type II inflammation, can you also identify, when possible, the genotypes studied in the references related to CRSwNP, as this may explain some of the difference reported in markers such as p53.
Author Response
Reviewer 2
I would like to thank the authors for all of there work on this important topic. This contribution helps fill in a gap in our understanding of the Inflamatory processes invloved in chronic rhinosinusitis, however there are a few areas within the paper that would bennifit from clarification and more information if possible.
Dear reviewer
We are very grateful for your review and all your comments. I hope that our supplement to the introduction and discussion according to your suggestions will increase the value of our paper
- line 38 suggest that chronic rhinosinusitis is an autoimmune disorder, and while there is significant evidence that autoimmune disorders contribute to the inflammatory processes involved, CRS id not considered an autoimmune disorder.
The theory of an autoimmune basis for CRS seems exciting and may guide further research on this topic. Shih at all. shows that chronic rhinosinusitis (CRS) with and without nasal polyps demonstrated a significant association with premorbid autoimmune diseases (i.e. ankylosing spondylitis, polymyositis, psoriasis, rheumatoid arthritis, sicca syndrome, and systemic lupus erythematosus or type 1 autoimmune pancreatitis). However, studies in the literature are limited.
Shih, LC., Hsieh, HH., Tsay, G.J. et al. Chronic rhinosinusitis and premorbid autoimmune diseases: a population-based case-control study. Sci Rep 10, 18635 (2020). ttps://doi.org/10.1038/s41598-020-75815-x,
Yoshikawa, T., Minaga, K., Hara, A., Sekai, I., Otsuka, Y., Takada, R., ... & Kudo, M. (2021). A unique profile of serum cytokines in type 1 autoimmune pancreatitis and chronic rhinosinusitis. Asian Pacific Journal of allergy and immunology.
- With the introduction of IG E monocolonal antibodies as a treatment for CRS, there has been significant research and discussion focused on distinguishing genotype from phenotype in Chronic Rhinosinusitis especially when considering medical therapy.
Throughout the paper you target the phenotype CRSwNP yet utilize genetic and molecular markers to define causes of inflammation involved. While Type II inflammation is most often involved in CRSwNP in western populations, it is the exact opposite in Asian Populations where type I inflammation is more typically involved in polyp formation.
- In line 86 you mention that you identified all patients had confirmed eosinophilic sinusitis, can you clarify that the genotype of each or the 10 patients with CRSwNP was type II inflammation, can you also identify, when possible, the genotypes studied in the references related to CRSwNP, as this may explain some of the difference reported in markers such as p53.
The Introduction section Is now supplemented with information about the phenotypes of chronic sinusitis:
CRS phenotypes have been defined as CRS with or without nasal polyposis, and subphenotypes include allergic fungal rhinosinusitis and CRS-related aspirin-exacerbated respiratory disease. Patients with CRS can be classified into three endotypes based on the presence of type 1, type 2, or type 3 inflammation. (Anna G. Staudacher, Anju T. Peters, Atsushi Kato, Whitney W. Stevens; Use of endotypes, phenotypes, and inflammatory markers to guide treatment decisions in chronic Rhinosinusitis, Annals of Allergy, Asthma & Immunology, Volume 124, Issue 4,2020,p.318-325,ISSN 1081-1206,https://doi.org/10.1016/j.anai.2020.01.013.)
Eosinophil count in nasal mucosa has been proved for genotyping and assessing disease severity. (Yao, Y., Xie, S., Yang, C. et al. Biomarkers in the evaluation and management of chronic rhinosinusitis with nasal polyposis. Eur Arch Otorhinolaryngol 274, 3559–3566 (2017). https://doi.org/10.1007/s00405-017-4547-2)
We added information about the number of eosinophils to the material and methods section,
Eosinophilic sinusitis in our work was determined by histological evaluation of the number of eosinophils in the visual field, which according to EPOS 2020 should be 10 or more in the visual field. (Fokkens WJ, Lund VJ, Hopkins C, et al. Executive summary of EPOS 2020 including integrated care pathways. Rhinology. 2020 Apr;58(2):82-111. DOI: 10.4193/rhin20.601. PMID: 32226949)
We have also completed the Introduction part with below content:
Tomassen et al divided patients with CRS on the basis of tissue immune markers in a phenotype-free approach. They identified patients with CRS associated with TH2- and eosinophil-driven inflammation, neutrophilic and proinflammatory cytokines,H17-TH22-related markers, and TH1, IFN-g markers. Their research focused on the relationship between endotypes and CRS phenotypes. CRSwNP endotype markers with nasal polyposis indicate the desirability of using biological medicines targeting these receptors (IgE, IL-4, IL-5 and IL-13), especially in the treatment of CRS with nasal polyps and asthma (IL-5). While significant progress has been made in characterizing endotypes, and phenotypes in CRS, additional studies are needed to determine how biomarkers could help physicians in individualized clinical treatment, and our work attempts to identify additional biomarkers of CRSwNP.
(Tomassen P.Vandeplas G.van Zele T.Cardell L.-O.Arebro J.Olze H.et al.Inflammatory endotypes of chronic rhinosinusitis based on cluster analysis of biomarkers. J Allergy Clin Immunol. 2016; 137: 1449-1456.e4)
Thank you very much for your valuable comments. We do hope our additional input clarifies and makes the article complementary

Reviewer 3 Report
Dear authors,
Thank you for submitting your manuscript to Biomedicines. Please kindly find below my remarks that I hope will increase the quality of the paper.
Line 37- Please revise the sources. There are four sources for one sentence.
Line 77 - I consider that the study group is too reduced to confirm the results.
Line 325 - The Conclusions section should summarise the practical impact of the results and how the current study can optimise future clinical protocols.
Best regards!
Author Response
Thank you for submitting your manuscript to Biomedicines. Please kindly find below my remarks that I hope will increase the quality of the paper.
Comment 1
Line 37- Please revise the sources. There are four sources for one sentence.
"Apoptosis plays an essential role in eliminating damaged cells and preventing uncontrolled cellular proliferation [1–4]."
Answer 1
The number of citation sources has been reduced in line 37, in accordance with the reviewer's comments .
Comment 2
Line 77 - I consider that the study group is too reduced to confirm the results.
Answer 2
In order to evaluate the test repeatability performance of the measurements, we used Stain’s formula.
The number of repetitions required to obtain significance (n) was measured for each assay. The t value at 9 degree degrees of freedom was taken form the Student distribution tables and was equal to 2.262. As a result, the calculated value of the minimum sample size (n) was lower than the assumed number of subjects used in the study (n=10), and the results were considered reproducible. The results of the calculations are presented in the Table below.
As authors, we agree that the group is small, but we hope the number of markers will compensate for this. We wanted the number of respondents to correspond to the control group. However, it is not easy to obtain histological material from healthy people. We think it will indicate further directions for developing research on CRS in NP.
- Line 325 - The Conclusions section should summarise the practical impact of the results and how the current study can optimise future clinical protocols.
We added a sentence in the conclusions section:
"Moreover, this is the first time the obtained studies showed increased levels of miR 203a-3p with decreased levels of miR 17-5p and miR-145-5p at the same time, suggesting that miR 203a-3p may be a potential therapeutic target for the treatment of CRSwNP. To a precision-medicine based approach for patient's treatment, a profound understanding of the pathology of this heterogeneous disease is crucial".
Thank you very much for your valuable comments.
We completed the introduction and method with additional information and added current references. We do hope our additional input clarifies and makes the article complementary.

Round 2
Reviewer 1 Report
The manuscript is well revised.
The quality of IHC in Figure 1 is still poor and not improved. Epithelial cells are exfoliated and damaged. They should be replaced with qualified ones. Even using AI techniques, quantification is impossible based on unfavorable materials.
Author Response
Dear Reviewer,
Once again, thank you for your comment. We appreciate all of the valuable indications to our manuscript.
We should not modify the images we obtained in this procedure due to the policy of Biomedicine (MDPI: https://www.mdpi.com/journal/biomedicines/instructions regarding image manipulation).
We performed serial repetitions of TUNEL in situ staining protocol using paraffin specimens. Although the obtained specimens were difficult to image, as mentioned before, we used all gathered images for quantification with ImageJ software. With regards to your suggestion, we have improved Figure 1, providing different images and photo shots of tissue specimens.
However, please also see the reference images obtained using the same protocol on paraffin-embedded tissues:
https://www.abcam.com/tunel-assay-kit-hrp-dab-ab206386.html
https://www.merckmillipore.com/PL/pl/product/ApopTag-Plus-Peroxidase-In-Situ-Apoptosis-Kit,MM_NF-S7101?ReferrerURL=https%3A%2F%2Fwww.google.com%2F
https://figshare.com/articles/figure/_a_LGP1_treatment_inhibited_the_apoptosis_in_mouse_kidney_tissue_TUNEL_staining_assay_scale_bar_200_956_m_/866894
https://www.semanticscholar.org/paper/Detection-of-apoptosis-in-paraffin-embedded-the-of-Dubsk%C3%A1-Matalov%C3%A1/56a4857a0e4c9cb384d2963fc25488811d9fdd38
Thank you for pointing out this problem.
Sincerely,
The authors
Reviewer 2 Report
I want to thank the authors for their response and the changes to the paper.
Author Response
We appreciate the Reviewer’s input in reviewing our manuscript and all constructive comments. Thanks to your feedback we could significantly improve our manuscript.
Best regards,
Authors
Reviewer 3 Report
Dear authors,
Thank you very much for providing the revised version of your manuscript.
The quality of the manuscript has been significantly improved.
I would kindly advise you to focus more on emphasising the clinical/practical impact of the results of your study in the Conclusions chapter.
Kind regards!
Author Response
Dear Reviewer,
We would like to thank you for the second revision. This feedback has raised some excellent points, and we sincerely appreciate these well-thought comments. We have improved the Conclusion chapter and added more detailed sentences about the clinical impact of the results of our study. The text has been checked and corrected. We appreciate the suggestions.
Best regards,
Authors